# NGLY1 as an Emerging Critical Modulator for Neurodevelopment and Pathogenesis in the Brain

**DOI:** 10.3390/ijms26199705

**Published:** 2025-10-06

**Authors:** Haiwei Zhang, Haipeng Xue, Yu-Chieh Wang, Ying Liu

**Affiliations:** 1Center for Translational Science, Florida International University, 11350 SW Village Pkwy, Port St. Lucie, FL 34987, USA; haizhang@fiu.edu (H.Z.); haxue@fiu.edu (H.X.); 2Robert Stempel College of Public Health and Social Work, Florida International University, Miami, FL 33199, USA; 3Applied Biomedical Science Institute, San Diego, CA 92127, USA; jack.wang@absinstitute.org; 4OrganoidPro LLC, San Diego, CA 92037, USA

**Keywords:** rare genetic disorders, NGLY1 deficiency, iPSCs, disease modeling, proteostasis

## Abstract

N-glycanase 1 (NGLY1) is a cytoplasmic glycoenzyme that removes N-linked glycans from misfolded glycoproteins. It plays an important role in the endoplasmic reticulum-associated degradation (ERAD) pathway in mammalian cells. NGLY1 dysfunction in humans causes NGLY1 deficiency as a rare autosomal recessive disorder that is characterized by neurodevelopmental delay, hypotonia, movement disorders, seizures, and multi-system involvement. In this review, we summarize recent advances in understanding the neural functions of NGLY1 and the neuropathological phenotypes associated with its deficiency. We discuss the molecular basis of NGLY1 deficiency in the central nervous system (CNS) and pathophysiological insights from animal and human induced pluripotent stem cell (iPSC)-based models. We also highlight emerging gene therapy approaches aimed at restoring NGLY1 activity and alleviating neurological symptoms.

## 1. Introduction

N-Glycanase 1 (also known as PNGase or human PNGase, hereafter referred to as NGLY1 for the protein, *NGLY1* for the human gene, and *Ngly1* for the rodent gene) is a cytoplasmic enzyme that catalyzes the deglycosylation of *N*-linked glycans from denatured glycoproteins, facilitating their proteasome-mediated degradation. NGLY1 is highly conserved across various eukaryotic organisms. The primary function of this enzyme enables the non-lysosomal hydrolysis of an N(4)-(acetyl-β-d-glucosaminyl)asparagine residue to yield N-acetyl-β-d-glucosaminyl-amine and a peptide containing an aspartic acid residue, ensuring the release of intact *N*-glycans from *N*-glycosylated proteins. Previous reports show that NGLY1 plays a pivotal role in the endoplasmic reticulum-associated degradation (ERAD) pathway, and its enzymatic activity is central to maintaining protein quality control and cellular homeostasis [1,2,3,4]. Consequently, any impairment or absence of NGLY1 function may significantly perturb normal cell functions and be implicated in several pathological conditions, including NGLY1-congenital disorder of deglycosylation (NGLY1-CDDG, also known as NGLY1 deficiency).

The recognition of NGLY1 deficiency as a distinct clinical diagnosis in humans is relatively recent. The first patient presenting with an autosomal recessive genetic disorder attributed to NGLY1 dysfunction was discovered in 2012 [5]. Once considered an ultra-rare genetic disorder, NGLY1 deficiency now has over 100 reported cases worldwide, with more than 60 clinically characterized, underscoring the growing recognition of its biological and clinical significance. Nonsense mutations causing premature termination of protein translation and small indel mutations leading to frameshift are found in patients with NGLY1 deficiency and their parents. Biallelic mutations in the *NGLY1* gene often result in the ablation of NGLY1 protein expression and a variety of systemic and neurological symptoms, including developmental delay, seizures, and motor dysfunction in probands. Although the phenotypes of NGLY1 deficiency are heterogeneous, the central nervous system (CNS) is a primary site of disease manifestations [6], highlighting the importance of NGLY1 in the development and normal function of neural cells, a role that remains to be fully elucidated. In addition, a better understanding of NGLY1 function in different cellular components forming central and peripheral neural networks is likely to generate valuable insights into the implications of dysregulated NGLY1 in the pathogenesis and disease progression of neurological abnormalities.

Recent reports using animal and human induced pluripotent stem cell (hiPSC)–derived organoid models for NGLY1 deficiency have revealed the potential impact of NGLY1 dysfunction on the CNS. Neurological defects, including motor impairment, abnormal locomotion, and reduced neuronal activity, are reported in *Ngly1*-knockout mice, rats, zebrafish, drosophila, as well as nematodes [7,8,9,10,11,12]. Moreover, the delivery of a functional copy of human *NGLY1* into the *Ngly1*-deficient rat brain alleviated motor and neurobehavioral deficits [13]. Using cerebral organoids developed from human pluripotent stem cells (hPSCs), our research team showed that NGLY1-null/mutant hPSCs had premature neuronal differentiation and gave rise to defective formation of SATB2-expressing upper-layer neurons compared to normal controls [14]. More recently, by characterizing NGLY1-deficient iPSC-midbrain organoids, Zheng’s group revealed a marked decrease in tyrosine hydroxylase expression, suggesting potential impairment of the dopaminergic pathway [15].

In this review, we discuss recent advances in NGLY1 research in the CNS and their clinical implications. We aim to highlight NGLY1 function under both normal and pathological conditions, thereby facilitating the discovery of novel therapies targeting its dysregulation in neurodegenerative diseases. Additionally, this review integrates new mechanistic insights from patients’ induced pluripotent stem cell (iPSC)-derived organoids, single-cell transcriptomics, and stress-response signaling to provide an expanded model of NGLY1 biology. Specifically, we emphasize (i) the emerging link between NGLY1-nuclear respiratory factor 1 (NRF)1 signaling and mitochondrial homeostasis, (ii) the discovery of immune-modulatory effects of NGLY1 on PD-1/PD-L1 regulation, and (iii) the differential impact of NGLY1 loss in human vs. rodent neurodevelopment, which explains why microcephaly could be prominent in patients but not in standard mouse models. By combining these mechanistic discoveries with translational updates, such as AAV9 gene-replacement studies and biomarker-guided clinical trial designs, our review provides forward-looking perspectives that help identify actionable therapeutic targets and experimental gaps not addressed in earlier publications.

## 2. NGLY1 Function in the CNS

### 2.1. Deglycosylation Activity of NGLY1 and Its Relevance to Proteostasis

The enzymatic activity of NGLY1 to strip high mannose-type *N*-glycans from misfolded glycoproteins is well characterized [16,17]. By catalyzing the hydrolysis of the amide bond between the innermost *N*-acetylglucosamine (GlcNAc) of an *N*-glycan conjugated to an asparagine (Asn) residue on a glycoprotein, NGLY1 releases the entire *N*-glycan and generates a de-*N*-glycosylated protein, where the *N*-glycosylated Asn residue is converted to an aspartic acid residue with a free 1-amino-GlcNAc-containing oligosaccharide.

The mammalian NGLY1 is a cytoplasmic PNGase which is distinct from other PNGases (e.g., glycoamidase/PNGase A and *N*-glycanase/PNGase F) found in plants and bacteria in several enzymatic properties, including the requirement of protein denaturation for activity and a neutral pH for optimal activity glycan cleavage. Another unique feature is that rodent Ngly1 and human NGLY1 have a carbohydrate-binding property [1,18]. The catalytic domain in the structures of Ngly1 and NGLY1 contains a core GlcNAc_2_-binding site [19,20]. There is also a PAW domain (a domain present in PNGases and other worm proteins) [21] reported to serve as a carbohydrate-binding domain for high mannose-type glycans [22] found at the C-terminus of either Ngly1 or NGLY1 protein. Interestingly, a PUB domain (a domain present in *P*NGase/*UB*A or *UB*X-containing proteins) [23] exists at the N-terminus of the enzyme and mediates the Protein–Protein Interactions of Ngly1/NGLY1 with other components in the ubiquitin-proteasome pathway [24].

The gene encoding cytoplasmic PNGase was first identified in yeast (*S. cerevisiae*) [25] and orthologs were later found in various eukaryotes, including mammals [25,26]. In mice, a 2.6 kb Ngly1 transcript is present in all tissues examined, with the highest expression in the testis [27]. Enzymatic activity is also detected in all mouse tissues tested, with the liver showing the highest overall activity [28,29,30,31]. As a cytoplasmic PNGase, NGLY1 has been reported to deglycosylate several ERAD substrates, although whether Ngly1-mediated deglycosylation plays an essential role in ERAD substrate degradation remains undetermined [29,30,31]. For example, although downregulation of Ngly1 reduces the deglycosylation of the TCR α subunit and MHC class I heavy chain, two well-known glycoproteins and ERAD substrates, their proteasome-mediated degradation was not affected [32,33,34]. In contrast, the degradation of EDEM1, a glycoprotein that targets misfolded proteins in the ER for degradation, appeared to be Ngly1-dependent and disrupted in response to Ngly1 inhibition [35]. Moreover, RTAΔm, a model ERAD substrate, accumulates in *Ngly1*-knockout mouse embryonic fibroblasts [36], indicating the critical role of NGLY1 in the degradation of RTAΔm. Overall, NGLY1 facilitates ERAD by accelerating the turnover of a subset of misfolded glycoproteins, but ERAD is not universally dependent on NGLY1, as several substrates (e.g., TCRα, MHC-I) are still degraded when NGLY1 is reduced, whereas others (e.g., EDEM1, RTAΔm) show clearer NGLY1 dependence or compensation by cytosolic ENGase, a cytosolic endo-β-N-acetylglucosaminidase, which can generate N-GlcNAc–modified proteins prone to aggregation (Figure 1).

### 2.2. Non-Enzymatic Function of NGLY1

Aside from the glycoenzyme function discussed earlier, NGLY1 may regulate gene transcription through an enzymatic activity-independent mechanism. For instance, *Ngly1*-deficient mouse embryonic fibroblasts show reduced mRNA and protein expression of the water channel gene Aqp1, likely accounting for their slower swelling in hypotonic solution [37]. NGLY1-deficiency patient fibroblasts and *NGLY1*-knockout human cells also showed reduced expression of AQP11 mRNA, supporting NGLY1 as a regulator for the expression of multiple aquaporin genes across species [37]. Notably, the exogenous expression of catalytically inactive NGLY1 (p.Cys309Ala) effectively restored Aqp1 expression in Ngly1-deficient mouse cells and their normal hypotonic lysis [37]. Further, the regulation of Aqp1 gene expression by transcription factors Creb1 and Atf1 was found to be disrupted in the Ngly1-deficient cells. In addition to aquaporins, NGLY1 has been shown to regulate the expression of the genes encoding proteasome subunits through modulating the glycosylation state and activity of transcription factor NFE2L1/NRF1 in response to proteasome inactivation [38]. Taken together, NGLY1 can contribute to the transcriptional regulation of gene expression, possibly through direct or indirect interactions with other transcription factors, which does not require the glycan-cleavage activity of this enzyme. While additional investigations are necessary, it is reasonable to consider NGLY1 as a versatile modulator that can regulate gene expression networks by orchestrating several transcription factors in different cell types under specific conditions (Figure 1).

### 2.3. Expression of NGLY1 in the CNS

Ngly1 expression is found throughout the entire brain in mice and rats. However, in Ngly1-deficient rats, different brain regions are affected to varying degrees, with neurodegeneration being particularly pronounced in the thalamus, spinal cord, and pons [8]. Additionally, the cerebellum was also affected in *Ngly1*-knockout C57BL/6J mice, which showed a decreased number of calbindin+ Purkinje cells along with motor deficits and gait abnormalities [39]. Similarly, the NGLY1 gene is ubiquitously expressed across human brain regions, though expression levels vary by region (https://www.proteinatlas.org/ENSG00000151092-NGLY1/brain (accessed on 22 July 2025). In NGLY1-deficiency patients, multiple structural and functional brain abnormalities have been observed [6,40]. Collectively, these data likely reflect region-specific expression and functional significance of NGLY1 in both animal models and human patients [8,39].

## 3. Neural Abnormalities Caused by NGLY1 Dysfunction

### 3.1. NGLY1 Gene Mutations Discovered in Humans

Many mutation types and spots in the *NGLY1* gene have been identified in patients with NGLY1 deficiency. Nonsense mutations that cause premature stop codons in the gene transcripts and subsequently lead to truncated and presumably non-functional proteins [4] are commonly seen. Indel mutations that result in a translational frameshift in the NGLY1 protein are also found but appear less frequently than point mutations [6,40,41,42]. The heterogeneity of the *NGLY1* gene mutations possibly contributes to the high variability of disease phenotypes in NGLY1-deficiency patients. It is known that NGLY1 deficiency leads to a variety of clinical presentations with different expressivity across different patients [6,40,43]. To date, more than 70 distinct pathogenic mutations in the *NGLY1* gene have been identified from slightly over 100 patients reported worldwide [43]. Because these mutations are distributed across the entire NGLY1 protein, each may uniquely affect its stability and/or activity, potentially contributing to the diverse types and severities of clinical presentations observed in patients. For instance, mutations that virtually ablate protein expression are likely to cause more prominent phenotypes in multiple organs, whereas missense mutations that partially interfere with NGLY1 enzymatic activity may give rise to milder phenotypes. *N*-glycoproteomics comparison of fibroblasts from NGLY1-deficiency patients and control individuals revealed a variety of NGLY1 substrates that were previously unrecognized [44]. The diverse interactions between different mutant variants of NGLY1 with distinct glycoproteomes in different cell types could add to the highly variable phenotypes in different NGLY1-deficiency patients. In addition, the impact on the epigenetic regulation of gene expression due to NGLY1 deficiency has not been well investigated. Thus, although NGLY1 deficiency is a monogenic disorder, it represents a complex, context-dependent pathological condition. The principal clinical features and their estimated frequencies, along with the phenotypic severity of NGLY1 variants, are summarized in Table 1.

### 3.2. Clinical Findings and Neurological Symptoms in NGLY1-Deficiency Patients

In 2012, the first patient with an *NGLY1* gene mutation was identified through whole-exome sequencing [5]. Since then, additional NGLY1 mutations have been discovered across several ethnic backgrounds [6,40,41,50]. While the number of identified NGLY1-deficiency patients remains small in the world, many clinical presentations, including developmental delay, motor dysfunction, microcephaly, epilepsy, mental impairment, sensory deficits, peripheral neuropathy, and gastrointestinal symptoms, are frequently observed [40,46,50] (Table 1). Notably, microcephaly occurs in ~47–70% of patients [40,45,46]. The motor symptoms include hypotonia and ataxia, and reduced lower limbs mobility and coordination [6,40]. Seizures are also commonly seen, with close to 58.6% of patients presenting with myoclonic and atonic seizures [47]. Although the disease presentations in other organs may become milder as the patients grow, their neurological symptoms appear to deteriorate and are accompanied by intellectual disability, leading to substantial challenges in learning and daily functioning (Figure 2). In addition, MRI and EEG also reveal brain structural changes and abnormal activities [6,40].

### 3.3. The Relevance of NGLY1 Dysregulation with Other Neuropathological Conditions

Beyond neurodevelopmental defects, NGLY1 dysfunction may also influence neurodegenerative processes. Recent work suggests that NGLY1 regulates tau seeding, aggregation, and turnover [51], raising the possibility of a broader role in disorders such as Alzheimer’s disease, Parkinson’s disease, amyotrophic lateral sclerosis, frontotemporal dementia, and Huntington’s disease, all of which share proteotoxic stress as a hallmark. However, no causal link between NGLY1 and these conditions has been established. Instead, current findings highlight a potential mechanistic connection. For example, the reported role of NGLY1 in tau regulation could position it as a possible modifier of neurodegenerative pathways, pending further validation in disease models and patient tissues [52]. Additionally, NGLY1 could also play a role in the peripheral nervous system (PNS), as peripheral neuropathy was observed in NGLY1-deficiency patients, and PNS defects have been reported in png-1-mutant roundworms [40,43,50,53].

### 3.4. The Difference of Defective NGLY1 on Neurodevelopment in Humans and Rodents

Microcephaly is frequently observed in NGLY1-deficiency patients [6,40,42]. While the rodent models of NGLY1 deficiency may show some neurological defects [7,8,13,39,54,55], microcephaly, at the level of gross anatomy, does not appear to be a phenotype commonly observed in NGLY1-deficient mice or rats. This discrepancy between human and rodent brains in response to NGLY1/Ngly1 deficiency may stem from fundamental differences in neurodevelopment between gyrencephalic brains (e.g., human and primate) and lissencephalic brains (e.g., rodent). It is known that a large number of cells present and highly proliferating in the outer subventricular zone of the developing primate brain do not exist in the rodent brain [56]. Compared with human brain development under specific gene dysregulation, rodent models with the homologous gene defect likely lack critical information about cell types unique to the developing human brain. Hence, it is reasonable to hypothesize that the neurogenesis and cell maturation defects caused by loss of NGLY1 function in humans could still be at least partially recapitulated in the developing Ngly1-deficient rodent brain. Nevertheless, Suzuki’s group has reported a lower brain weight of *Ngly1*-knockout rats compared to wild-type counterparts, supporting a microcephaly-related phenotype [8].

Rodent models have been indispensable for elucidating the fundamental biology of NGLY1, modeling disease mechanisms, and identifying potential therapeutics. However, they do not always fully recapitulate the human conditions. Anatomical divergence suggests that rodent efficacy and safety data may underestimate human neurodevelopmental vulnerability. Therefore, confirmatory studies in higher-order systems, such as human iPSC-derived organoids, humanized animal models, or non-human primates, are critical for improving clinical translation. Together, these complementary models will help ensure that early-phase exploratory trials remain relevant to the human disease trajectory.

### 3.5. Abnormal Neurodevelopment Due to NGLY1 Deficiency Modeled by Human Neural Organoids

Given the limitations of rodent models in studying NGLY1-related neurodevelopmental defects, human cell-based models offer a highly relevant approach to investigate how NGLY1 deficiency impacts brain development. Human induced pluripotent stem cells (hiPSCs) and hiPSC-developed brain organoids have been shown to recapitulate the early development of the human brain and model neurodevelopmental disorders [57]. We and others have generated NGLY1-deficiency patient iPSCs and NGLY1 knockout human embryonic stem cells (hESCs), and to obtain cerebral organoids (COs) for modeling NGLY1-deficiency [14,15,58,59,60,61,62,63]. For example, we found that NGLY1-deficient COs exhibited a marked reduction in SATB2-expressing upper-layer neurons, along with impaired STAT3 and HES1 signaling essential for sustaining neural stem cell (NSC) vitality and function during brain development. Single-cell transcriptomic analysis further indicated a premature neuronal differentiation in NGLY1-deficient COs [14]. Also, the *IGFBP2* and *ID4* genes, known for their important roles in regulating NSC properties and differentiation [64,65,66,67], were shown to be significantly downregulated in NGLY1-deficient organoid [14]. In contrast to Id4 expression in the mouse embryonic brain that peaks at E11.5 and dissipates, ID4 expression in the human embryonic brain persists onward from Carnegie stages (CS)13 [68]. This prolonged expression of ID4 in the developing human brain highlights the unique requirement of ID4 expression for proper human brain development. The disruption of ID4 expression discovered in the NGLY1-deficient human COs further supports the importance of NGLY1 in the regulation of human neurogenesis and may also help explain why NGLY1/Ngly1 deficiency appears to have a differential impact on brain development. Additionally, midbrain organoids derived from *NGLY1*-deficiency patient iPSCs also show impaired neuronal and astrocytic differentiation, reduced GABAergic signaling, and diminished expression of dopaminergic neuron markers [15]. Taken together, hiPSC-organoids represent an important tool for modeling NGLY1-deficiency.

### 3.6. Defective NGLY1-Induced Dysregulation of Cell Stress Responses and Immune Signaling

In addition to its role in neurodevelopmental signaling, NGLY1 is also implicated in regulating stress responses and immune signaling. For instance, we showed that NGLY1-deficient COs have significantly higher susceptibility to proteasome inhibitor-induced proteotoxicity, oxidative stress, and thapsigargin-induced ER stress, but appeared to have higher tolerability to the challenge of cisplatin-induced genotoxicity and glucose deprivation, while exogenous NGLY1 expression restored cell tolerance to both proteasome inhibition and hydrogen peroxide [14]. Consistently, a recent report showed that the restoration of NGLY1 expression by gene editing reduces ER stress, oxidative stress, and autophagy in NSCs differentiated from NGLY1-deficiency patient-derived hiPSCs [60]. In both human and mouse fibroblasts that have lost NGLY1/Ngly1 function, the chronic activation of cGAS–STING and MDA5–MAVS pathways and the elevated expression of interferon-stimulated genes were observed [54]. The aberrant stress-response and immune-regulatory signaling activation were linked to the reduced proteasome activity and impaired mitophagy and mitochondrial homeostasis as a consequence of disrupted transcriptional activity of NRF1/Nrf1 due to NGLY1/Ngly1 deficiency [54]. Similarly, the significant role of NGLY1 in the regulation of stress and immune signaling was shown in human melanoma cells, where the hyperexpression of NGLY1 was shown to be critical for the cancer cells to limit cell death [69]. A more recent discovery of NGLY1 activity coupling to the regulation of PD-1 deglycosylation and degradation in T cells but not affecting glycosylated PD-L1 stability [70] further supports that NGLY1 could modulate the immune system in context- and cell type-dependent manners. With these findings, it is reasonable to speculate that NGLY1 deficiency may cause abnormal immunogenicity or chronic inflammatory responses. However, unlike the unequivocal activation of type-I and type-III interferon expression in human melanoma cells subjected to NGLY1 inactivation, a similar response appears to be absent in neural cells developed in NGLY1-deleted human COs compared to their isogenic, NGLY1-functional counterparts [14].

Modeling NGLY1 deficiency with hiPSC-derived COs presents both advantages and limitations. COs reveal core NGLY1 phenotypes, including premature neurogenesis, loss of SATB2+ upper-layer neurons, and disruption of STAT3/HES1, with down-regulation of IGFBP2/ID4. Patient-derived midbrain organoids also show reduced dopaminergic markers. These models are stress-sensitive (proteasome inhibition, oxidative, and ER stress) and can be rescued by NGLY1 re-expression. Despite the significant discoveries made in the NGLY-deficiency organoids, standard organoid models of early fetal stages lack full vascular and immune components, limiting inferences about later maturation and neuroinflammation. Priority next steps include testing ENGase-dependent N-GlcNAc aggregation and its reversibility (ENGase inhibition) in neural contexts, moving to vascularized organoids and microglia-containing assembloids, extending culture time, and aligning organoid readouts with translational biomarkers such as GlcNAc-Asn.

For immune signaling, current evidence shows that NGLY1 loss activates cGAS–STING/MDA5–MAVS and interferon programs in fibroblasts via impaired NRF1-linked proteostasis/mitochondria, whereas neural cells in cerebral organoids do not display the same IFN activation at baseline, a gap likely reflecting cell-type thresholds and missing glia/immune crosstalk. Moreover, NGLY1 promotes PD-1 deglycosylation and degradation in T cells (without destabilizing PD-L1), suggesting context-dependent immunomodulation whose CNS relevance remains to be tested. A more mechanistic program should probe triggered conditions (e.g., proteasome/oxidative stress where organoids are vulnerable), quantify innate-immunity readouts (cGAMP, STING phosphorylation, ISG proteins), and ask whether NRF1 restoration or proteostasis modulators normalize mitochondrial/IFN phenotypes, ideally in neuron–microglia (with or without T-cell) co-cultures that can reveal PD-1/PD-L1 pathway consequences in neural circuits. In summary, although the direct evidence in support of NGLY1 dysfunction-induced neuroinflammation and its significance for the pathogenesis of neurological diseases remains to be established, testing the possible links between NGLY1 dysregulation and neurodegenerative disorders in further optimized systems may lead to critical discoveries that facilitate the development of novel therapies for better managing patients affected by neurodegeneration.

## 4. Development of Potential Therapeutic Strategies and Clinical Translation

The development of effective therapeutic strategies for NGLY1 deficiency has gained momentum in recent years. Recent studies have shown the feasibility of using AAV9 vectors to deliver functional NGLY1 genes into animal models, leading to the restoration of enzymatic activity and significant improvements in motor functions [13,71,72]. This approach underscores the promise of gene therapy as a viable treatment strategy for NGLY1 deficiency, particularly when administered early in the disease course. While AAV9-mediated gene replacement and CRISPR/Cas9 genome correction offer compelling proof-of-concept for durable restoration of NGLY1 activity, several challenges must be acknowledged. First, host immune responses, including neutralizing antibodies to AAV9 and innate inflammatory activation, can limit transduction efficiency and trigger adverse events, particularly when vectors are delivered systemically to reach the CNS. Second, achieving broad and uniform CNS distribution is also challenging. Intrathecal or intracerebroventricular routes improve brain exposure but add procedural complexity and risk, while systemic delivery often yields heterogeneous expression. Lastly, even when efficient delivery is achieved, maintaining long-term transgene expression is not guaranteed; episomal AAV genomes can be lost over time or subjected to epigenetic silencing, raising concerns for a lifelong disorder that may require re-dosing. CRISPR-based editing introduces additional uncertainties such as off-target cleavage, incomplete correction, and potential mosaicism in neuronal populations.

To complement these gene-directed approaches, several pharmacological or small-molecule strategies are being explored. Proteostasis modulators, including NRF1 activators and proteasome enhancers, may help compensate for defective deglycosylation by stabilizing protein quality-control networks. Antioxidants and stress-response regulators have reduced endoplasmic reticulum and oxidative stress in NGLY1-deficient human cerebral organoids, suggesting a potential to ameliorate downstream pathology even without restoring enzyme activity. In addition, high-throughput iPSC-based drug screening is identifying small molecules that modulate NGLY1-dependent pathways and may serve as either stand-alone therapies or adjuvants to gene editing. Together, these complementary approaches provide a diversified therapeutic landscape that balances the transformative potential of gene therapy with the practicality and incremental benefits of pharmacological intervention. Table 2 provides a summary of potential therapeutic and diagnostic strategies under investigation.

Proof-of-concept for AAV9 gene replacement and CRISPR/Cas9 correction is strong in rodents and patient-derived cells, with restoration of NGLY1 activity and functional improvement after CNS-directed or systemic AAV9 delivery, as well as successful gene correction in iPSC models. Nonetheless, clinical translation will require caution in three key areas: safety, delivery, and durability. Safety considerations include vector-related toxicity, potential immune activation, and off-target gene-editing effects. For delivery, the challenge is to achieve broad and sustained expression across heterogeneous brain regions, while preclinical work must carefully weigh trade-offs between intracerebroventricular/intrathecal and systemic routes. With respect to durability and re-dosing, strategies must ensure long-term expression in the context of a lifelong disorder. From a regulatory perspective, progress will depend on (i) rigorous biodistribution and potency data in relevant CNS models, (ii) trial designs that prioritize early intervention, and (iii) incorporation of surrogate or response biomarkers, particularly GlcNAc-Asn, to monitor pharmacodynamic effects in small patient cohorts. In parallel, a complementary small-molecule strategy should be advanced, using iPSC-based screening to identify agents that modulate NRF1, enhance proteostasis, or mitigate cellular stress. Such agents could be developed as monotherapies or as adjuncts to gene therapy, diversifying risk while gene-based approaches mature. Taken together, this tempered outlook balances optimism with pragmatic steps to accelerate translation, reduce risk, and ensure that forthcoming trials are interpretable and clinically meaningful.

The design of clinical trials for NGLY1 deficiency presents unique challenges. Early intervention is critical, as evidence suggests that the timing of therapeutic administration significantly influences outcomes [39]. Clinical trial designs must, therefore, prioritize early diagnosis and treatment initiation, potentially incorporating biomarkers for monitoring therapeutic efficacy. The identification of surrogate endpoints, such as the levels of specific biomarkers like GlcNAc-Asn, which correlate with NGLY1 activity, could facilitate the evaluation of treatment responses in clinical settings [74]. However, although GlcNAc-Asn has emerged as a promising biomarker for disease activity, the clinical diagnosis of NGLY1 deficiency remains difficult. The extreme rarity and heterogeneous presentation of NGLY1 deficiency, ranging from developmental delay to hepatic abnormalities, often lead to delayed recognition or misdiagnosis. Standard metabolic screening panels do not currently detect NGLY1 deficiency, and confirmatory genetic testing requires either targeted NGLY1 sequencing or whole exome or whole genome sequencing, which may not be available in all scenarios.

Nonetheless, early detection remains critical, as pre-symptomatic intervention can maximize the benefits of emerging gene and pharmacological therapies. Mass spectrometry using dried blood spots has enabled the detection of key metabolites relevant to inherited metabolic disorders [76]. Meanwhile, advances in next-generation sequencing have expanded the clinical use of exome and whole-genome sequencing for diagnosing genetic diseases with complex presentations. These approaches may also facilitate early detection of GlcNAc-Asn or *NGLY1* gene mutation in the future. However, implementation will require validation of assay specificity, cost-effectiveness assessments, and the development of clear treatment algorithms for identified infants.

Because NGLY1 deficiency is a lifelong, multisystem disorder, patient-reported outcomes and quality-of-life measures are essential for evaluating the real-world impact of treatment. Traditional clinical endpoints may not fully capture the burden of disease or the benefits of therapy, particularly in rare disorders, including NGLY1 deficiency, where improvements may be modest but highly meaningful to patients and families. Rare disease trials often employ validated instruments such as the pediatric quality of life inventory, which assesses physical, emotional, social, and school functioning; the caregiver impact questionnaire, which measures caregiver stress and daily challenges, and the clinical global impression of change, which provides an overall assessment of perceived clinical benefit. Additional tools, including the disease-specific modules adapted for developmental delay and neural disability, may further strengthen the evaluation framework. Incorporating these measures into future NGLY1 clinical studies will ensure that therapeutic efficacy is assessed not only through biochemical or neurodevelopmental endpoints but also through meaningful improvements in daily functioning, comfort, independence, and family well-being. This holistic approach aligns with current best practices in rare disease research, where patient and caregiver perspectives are increasingly recognized as critical to guiding regulatory approval and healthcare decision-making [75,77].

Clinical trials for rare genetic disorders have advanced significantly, particularly in gene therapy and personalized medicine, though challenges in accessibility and regulatory pathways remain. Advances in genomics, including next-generation and whole-genome sequencing, have accelerated both diagnosis and treatment development [78,79]. Recent trials increasingly incorporate CRISPR-Cas9 gene editing, RNA-based therapies, and small-molecule drugs. In 2025, the U.S. FDA introduced Rare Disease Evidence Principles to expedite approvals for ultra-rare genetic conditions, emphasizing flexibility in trial design given small patient populations. Complementary efforts, such as the Ultra-rare Gene-based Therapy Network, provide funding and infrastructure to advance gene therapy research for ultra-rare neurological disorders. The clinical trial development of NGLY1 deficiency will benefit from these resources, as they provide not only funding and infrastructure but also access to specialized expertise, standardized protocols, and regulatory guidance. Leveraging these initiatives can help overcome barriers posed by small patient populations, accelerate the transition from preclinical findings to human trials, and ensure that study designs are both feasible and scientifically rigorous. Together, these resources create an ecosystem that supports the advancement of safe, effective, and patient-centered therapies for NGLY1 deficiency.

In summary, the development of novel therapeutic strategies for NGLY1 deficiency is progressing rapidly. The integration of these approaches into clinical trial designs, with a focus on early intervention and the use of biomarkers, will be crucial in translating these promising therapies from bench to bedside. As research continues to evolve, there is hope that effective treatments will soon be available for patients suffering from this debilitating condition, significantly improving their quality of life and clinical outcomes.

## 5. Conclusions

NGLY1 deficiency causes profound CNS phenotypes. Studies using animal models and patient iPSCs have provided insights into the pathogenic mechanisms of NGLY1 deficiency. Additionally, genetic studies on NGLY1-deficiency patients have revealed a complicated genetic map. Future research can focus on detailed molecular characterization of NGLY1 deficiency. Importantly, hiPSC-derived brain organoids provide a powerful platform to study how NGLY1 deficiency disrupts neurogenesis, revealing alterations in neural stem cell signaling, premature differentiation, and loss of key neuronal populations. Collectively, organoid studies underscore the critical role of NGLY1 in human brain maturation and establish hiPSC-derived systems as indispensable for mechanistic and therapeutic exploration [14,15,58,59,60,61,62,63]. NGLY1 deficiency clearly perturbs proteostasis and neurodevelopment, but decisive CNS mechanisms remain unresolved. ERAD is not universally dependent on NGLY1, as some substrates are degraded without it, whereas others show clearer NGLY1 dependence or ENGase compensation that produces N-GlcNAc aggregates. Beyond catalysis, NGLY1 exerts non-enzymatic functions, including aquaporin transcription (rescued by a catalytic-dead allele) and regulation of NRF1-driven proteasome/mitochondrial programs. How these roles partition across neural lineages is unclear. With >70 pathogenic variants and marked phenotypic heterogeneity, potential separation-of-function classes are possible but unmapped. Priorities now are cell-type-resolved N-glycoproteomics in human neural models, genetic dissection of the NGLY1-ENGase axis, allele-specific rescue (catalytic-dead vs. wild-type) to parse catalytic vs. regulatory roles, and structure-guided variant stratification in isogenic neural cells linked to patient phenotypes; these advances will guide when to prioritize enzyme restoration versus proteostasis or NRF1 modulation and will enable variant-informed biomarkers (e.g., GlcNAc-Asn) in clinical trials. Insights obtained from these studies will reveal not only the etiology of the disorder but also new targets for therapies. Novel therapeutic approaches, such as advanced gene editing, as well as pharmacological interventions, have the potential to alter the course of disease. While holding a great promise, these strategies need to be validated in clinical trials in order to propagate their safety and efficacy within affected populations. Moreover, early recognition is key to effective care for individuals with NGLY1 deficiency. Adopting large-scale screening programs and the new genetic testing technologies can help in the early identification of affected cases.

## Figures and Tables

**Figure 1 ijms-26-09705-f001:**
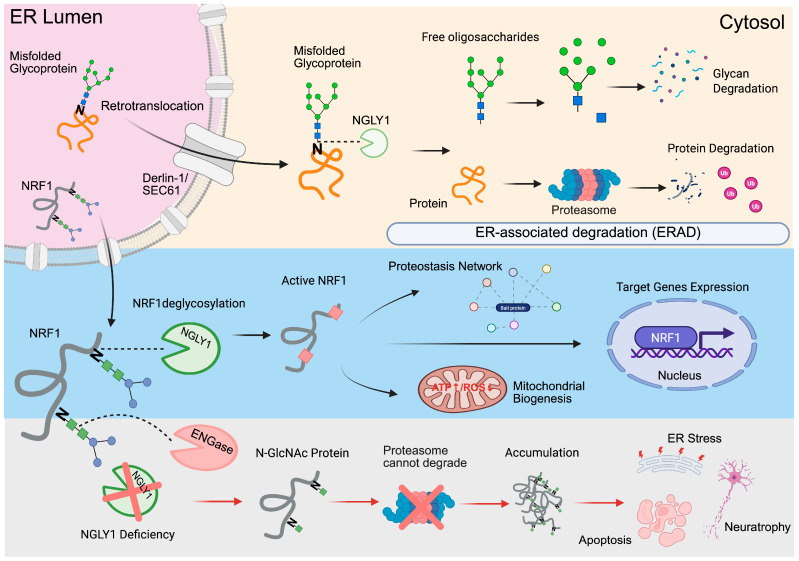
Role of NGLY1 in ER-associated degradation (ERAD) and neuronal homeostasis. The upper panel illustrates the ERAD pathway, where misfolded glycoproteins undergo retrotranslocation from the ER lumen to the cytosol, followed by deglycosylation by NGLY1 and subsequent degradation via the proteasome. The glycan chains are independently catabolized. The middle panel shows NGLY1-mediated deglycosylation of NRF1, enabling its activation and translocation to the nucleus, where it induces genes involved in proteostasis and mitochondrial function. The bottom panel depicts the pathological consequences of NGLY1 deficiency: ENGase processes the misfolded proteins into N-GlcNAc forms that accumulate due to proteasome resistance, leading to ER stress, apoptosis, and neurodegeneration. ERAD may proceed in an NGLY1-independent manner with ENGase-mediated deglycosylation for certain substrates, while other substrates exhibit partial NGLY1 dependence; thus, NGLY1 acts as a facilitator rather than a universal requirement for ERAD. ENGase, endo-beta-N-acetylglucosaminidase, a cytosolic enzyme that hydrolyzes mannose-modified peptides and proteins, generating free oligosaccharides. NRF1, nuclear respiratory factor 1, a transcription factor regulating mitochondrial function, proteasome activity, and stress responses, is essential for cellular homeostasis and adaptation. Created in BioRender. Zhang, H. (2025) https://app.biorender.com/profile/template/details/t-68e130204289dc9ed52224c8-role-of-ngly1-in-er-associated-degradation-erad-and-neuronal.

**Figure 2 ijms-26-09705-f002:**
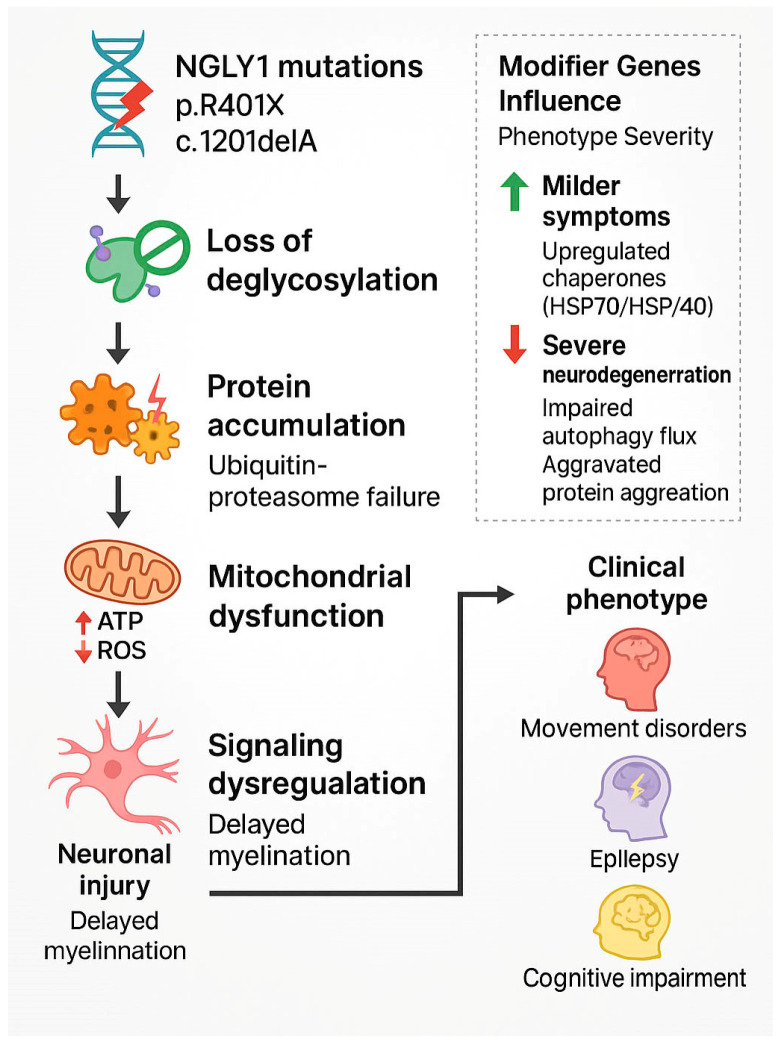
Neurological phenotypic cascade in NGLY1 deficiency. This scheme illustrates a likely process by which NGLY1 gene mutations lead to clinical neurological symptoms. Mutations in NGLY1 (e.g., p.R401X, c.1201delA) cause a loss of NGLY1 enzyme activity and defective deglycosylation. This leads to protein buildup, mitochondrial problems, and disrupted signaling, ultimately resulting in neuronal damage and developmental issues. Clinical presentations include movement disorders, epilepsy, and cognitive problems. A side panel emphasizes the role of modifier genes, which may affect disease severity through pathways such as heat shock factor 1 (HSF1)-driven chaperone upregulation or faulty autophagy due to autophagy-related gene 5 (ATG5) deficiency. ATP, adenosine triphosphate; HSP, heat shock protein. Note that p.R401X (c.1201delA) illustrated in this figure is one of the most common mutations found in NGLY1 deficiency patients. It arises from the deletion of an adenosine at nucleotide no. 1201, converting the codon for arginine at position 401 into a stop codon and causing premature termination of the NGLY1 protein. Created using Adobe Illustrator 2025.

**Table 1 ijms-26-09705-t001:** Summary of clinical features and associated genetic variants in NGLY1 deficiency.

Clinical Feature	Prevalence (Patient Count)	Variant Types	Severity	References
Motor/speech delay	>97% (~36/37)	Frameshift, missense	Moderate to severe	[40,45,46]
Epilepsy (myoclonic/atonic)	≈58.6% (~17/29)	Nonsense, missense	Moderate	[47]
Motor dysfunction (ataxia, hypotonia)	Common (>44/47)	Frameshift, premature stop	Severe	[40,45,46]
Intellectual disability	Highly variable (~9/12)	Various	Mild to severe	[6]
Microcephaly	≈47% (~22/47)	Various	Moderate	[40,45,46]
Hepatic abnormalities	≈65% (~30/46)	No specific variant association	Mild	[40,41,45,46]
Alacrima (absent tear production)	≈86.6% (~39/45)	Premature stop	Mild	[6,40,41,46]
Orthopedic phenotype	≈80% (~23/29)	Various	Mild	[48,49]

**Table 2 ijms-26-09705-t002:** Summary of potential therapeutic or diagnostic strategies in NGLY1 deficiency.

Strategy	Mechanism of Action	Delivery Method/Model	Current Development Stage	References
AAV9-mediated gene therapy	Restores *NGLY1* gene expression in CNS	Intracerebroventricular or systemic AAV9 injection in rats/mice	Advanced preclinical multiple rodent efficacy studies; IND-enabling work underway	[13,39,71,72]
CRISPR/Cas9 gene editing	Genome correction of pathogenic *NGLY1* mutations	Viral or non-viral vectors in human iPSCs	Preclinical (cell models); proof-of-concept correction in patient iPSCs	[14,73]
iPSC-based drug screening	High-throughput identification of phenotype modifiers	Human patient-derived iPSCs	Discovery stage; ongoing high-throughput screens	[14,15,60,63]
Biomarker-guided monitoring	Use GlcNAc-Asn levels to track disease activity and therapeutic response	Rat models, patient blood/CSF samples	Early clinical utility validated in patient samples; under consideration for trial endpoints	[71,74,75]

## Data Availability

No new data were created or analyzed in this study. Data sharing is not applicable to this article.

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
