# Peer review of "NGLY1 as an Emerging Critical Modulator for Neurodevelopment and Pathogenesis in the Brain"

_ijms, 2025, doi:10.3390/ijms26199705_

Round 1

Reviewer 1 Report

Comments and Suggestions for Authors

Overall Assessment

The manuscript provides a timely and comprehensive review of the role of NGLY1 in neurodevelopment and neurological disorders. It is well-structured, covers both enzymatic and non-enzymatic functions of NGLY1, and integrates evidence from animal models, iPSC-derived organoids, and therapeutic approaches. The topic is highly relevant for the readership of IJMS.

However, some areas require clarification, more balanced discussion, and additional references to strengthen the scientific rigor and clinical translational aspects.

Major Comments

  1. Scope and Novelty

    • The review adequately summarizes existing literature, but it would benefit from a clearer emphasis on novel insights. At present, the manuscript reads as a broad summary. Authors should highlight where their review provides unique perspectives compared to recent publications .

  2. Therapeutic Strategies Section (Section 4)

    • The discussion of gene therapy (AAV9, CRISPR) is promising but somewhat descriptive.

      • Please expand on the limitations and risks (e.g., immune responses, delivery challenges to CNS, long-term expression stability).

      • Incorporate more balanced perspectives, including potential pharmacological or small-molecule approaches in addition to gene editing.

  3. Clinical Translation

    • The review highlights biomarkers (e.g., GlcNAc-Asn) but does not adequately discuss diagnostic challenges. Early detection and newborn screening possibilities should be elaborated.

    • Patient-reported outcomes and quality-of-life measures are mentioned briefly—consider expanding with examples or frameworks used in rare disease trials.

  4. Comparative Models

    • The differences between human and rodent phenotypes (microcephaly discrepancy) are well described. However, the implications for translational validity of rodent models should be discussed more critically. How might reliance on rodents affect therapeutic development timelines?

  5. Figures and Tables

    • Figure 1 and Figure 2 are useful but would benefit from more explanatory legends (e.g., acronyms like ENGase, NRF1 should be spelled out).

    • Table 1: Include approximate patient numbers or references for prevalence rates to give stronger clinical context.

    • Table 2: Add a column for current development stage (preclinical, early-phase clinical, etc.) to give readers a sense of progress.

      Minor Comments

      1. Writing and Clarity

        • The manuscript is generally well written, but several sentences are lengthy and could be tightened for readability.

        • Example: Lines 21–24 (“We discuss the molecular basis … alleviate neurological symptoms”) could be split for clarity.

      2. References

        • Ensure reference formatting is consistent (some entries miss volume/issue formatting).

        • More recent papers (2023–2025) on clinical trial designs in rare genetic disorders could be cited for context.

      3. Conflict of Interest

        • One author is employed by a company (OrganoidPro LLC). While disclosed, it may be appropriate to clarify whether this affiliation had any role in data interpretation, especially regarding organoid models.

      4. Terminology

        • Use consistent terminology: “NGLY1 deficiency” vs. “NGLY1-CDDG.” Consider unifying to avoid reader confusion.

Author Response

Reviewer 1

Overall Assessment

The manuscript provides a timely and comprehensive review of the role of NGLY1 in neurodevelopment and neurological disorders. It is well-structured, covers both enzymatic and non-enzymatic functions of NGLY1, and integrates evidence from animal models, iPSC-derived organoids, and therapeutic approaches. The topic is highly relevant for the readership of IJMS.

However, some areas require clarification, more balanced discussion, and additional references to strengthen the scientific rigor and clinical translational aspects.

Major Comments

  1. Scope and Novelty
    • The review adequately summarizes existing literature, but it would benefit from a clearer emphasis on novel insights. At present, the manuscript reads as a broad summary. Authors should highlight where their review provides unique perspectives compared to recent publications .

Responses: Thank the reviewer for the valuable suggestion. We have added the following in Section 1 “Introduction” section to highlight our unique perspectives.

Unlike previous reviews that mainly describe the clinical features of NGLY1 deficiency or describe its role in ER-associated degradation, our manuscript integrates new mechanistic insights from patients’ induced pluripotent stem cell (iPSC)-derived organoids, single-cell transcriptomics, and stress-response signaling to provide an expanded model of NGLY1 biology. Specifically, we emphasize (i) the emerging link between NGLY1-nuclear respiratory factor 1 (NRF)1 signaling and mitochondrial homeostasis, (ii) the discovery of immune-modulatory effects of NGLY1 on PD-1/PD-L1 regulation, and (iii) the differential impact of NGLY1 loss in human vs. rodent neurodevelopment, which explains why microcephaly could be prominent in patients but not in standard mouse models. By combining these mechanistic discoveries with translational updates, such as AAV9 gene-replacement studies and biomarker-guided clinical trial designs, our review provides forward-looking perspectives that help identify actionable therapeutic targets and experimental gaps not ad-dressed in earlier publications.

Comments:

  1. Therapeutic Strategies Section (Section 4)
    • The discussion of gene therapy (AAV9, CRISPR) is promising but somewhat descriptive.
      • Please expand on the limitations and risks (e.g., immune responses, delivery challenges to CNS, long-term expression stability).
      • Incorporate more balanced perspectives, including potential pharmacological or small-molecule approaches in addition to gene editing.

Responses: Thank the reviewer for the advice. We have added the following to Section 4 of the manuscript to provide more discussion on therapeutic strategies.

While AAV9-mediated gene replacement and CRISPR/Cas9 genome correction offer compelling proof-of-concept for durable restoration of NGLY1 activity, several challenges must be acknowledged. Host immune responses, including neutralizing antibodies to AAV9 and innate inflammatory activation, can limit transduction efficiency and trigger adverse events, particularly when vectors are delivered systemically to reach the CNS. Achieving broad and uniform CNS distribution is also challenging. Intrathecal or intracerebroventricular routes improve brain exposure but add procedural complexity and risk, while systemic delivery often yields heterogeneous expression. Even when efficient delivery is achieved, maintaining long-term transgene expression is not guaranteed; episomal AAV genomes can be lost over time or subjected to epigenetic silencing, raising concerns for a lifelong disorder that may require re-dosing. CRISPR-based editing introduces additional uncertainties such as off-target cleavage, incomplete correction, and potential mosaicism in neuronal populations.

To complement these gene-directed approaches, several pharmacological or small-molecule strategies are being explored. Proteostasis modulators, including NRF1 activators and proteasome enhancers, may help compensate for defective deglycosylation by stabilizing protein quality-control networks. Antioxidants and stress-response regulators have reduced endoplasmic reticulum and oxidative stress in NGLY1-deficient human cerebral organoids, suggesting a potential to ameliorate downstream pathology even without restoring enzyme activity. In addition, high-throughput iPSC-based drug screening is identifying small molecules that modulate NGLY1-dependent pathways and may serve as either stand-alone therapies or adjuvants to gene editing. Together, these complementary approaches provide a diversified therapeutic landscape that balances the transformative potential of gene therapy with the practicality and incremental benefits of pharmacological intervention.

Comments:

  1. Clinical Translation
    • The review highlights biomarkers (e.g., GlcNAc-Asn) but does not adequately discuss diagnostic challenges. Early detection and newborn screening possibilities should be elaborated.
    • Patient-reported outcomes and quality-of-life measures are mentioned briefly—consider expanding with examples or frameworks used in rare disease trials.

Responses: Thank the reviewer for the advice. We have added the following to Section 4 to highlight early detection and potential therapeutics.

Although GlcNAc-Asn has emerged as a promising biomarker for disease activity, the clinical diagnosis of NGLY1 deficiency remains difficult. The extreme rarity and heterogeneous presentation of NGLY1 deficiency, ranging from developmental delay to hepatic abnormalities, often leads to delayed recognition or misdiagnosis. Standard metabolic screening panels do not currently detect NGLY1 deficiency, and confirmatory genetic testing requires either targeted NGLY1 sequencing or whole exome or whole genome sequencing, which may not be available in all scenarios.

Nonetheless, early detection remains critical, as pre-symptomatic intervention can maximize the benefits of emerging gene and pharmacological therapies. Mass spectrometry using dried blood spots has enabled detection of key metabolites relevant to inherited metabolic disorders (Moat et al, 2020). Meanwhile, advances in next-generation sequencing have expanded the clinical use of exome and whole-genome sequencing for diagnosing genetic diseases with complex presentations. These approaches may also facilitate early detection of GlcNAc-Asn or NGLY1 gene mutation in the future. However, implementation will require validation of assay specificity, cost-effectiveness assessments, and the development of clear treatment algorithms for identified infants.

Because NGLY1 deficiency is a lifelong, multisystem disorder, patient-reported outcomes and quality-of-life measures are essential for evaluating the real-world im-pact of treatment. Traditional clinical endpoints may not fully capture the burden of disease or the benefits of therapy, particularly in rare disorders including NGLY1 deficiency, where improvements may be modest but highly meaningful to patients and families. Rare disease trials often employ validated instruments such as the pediatric quality of life inventory, which assesses physical, emotional, social, and school functioning, the caregiver impact questionnaire, which measures caregiver stress and daily challenges, and the clinical global impression of change, which provides an overall assessment of perceived clinical benefit. Additional tools, including the disease-specific modules adapted for developmental delay and neural disability, may further strengthen the evaluation framework. Incorporating these measures into future NGLY1 clinical studies will ensure that therapeutic efficacy is assessed not only through bio-chemical or neurodevelopmental endpoints but also through meaningful improvements in daily functioning, comfort, independence, and family well-being. This holistic approach aligns with current best practices in rare disease research, where patient and caregiver perspectives are increasingly recognized as critical to guiding regulatory approval and healthcare decision-making (Tong et al, 2023; Slade et al,2018).

Comments:

  1. Comparative Models
    • The differences between human and rodent phenotypes (microcephaly discrepancy) are well described. However, the implications for translational validity of rodent models should be discussed more critically. How might reliance on rodents affect therapeutic development timelines?

Responses: Thank the reviewer for pointing this out. We have clarified the advantages and limitations of modeling NGLY1 deficiency in rodent models in Section 3.4 of the manuscript.

Rodent models have been indispensable for elucidating the basic biology of NGLY1, modeling disease mechanisms, and identifying potential therapeutics. However, they do not always fully recapitulate the human condition. Anatomical divergence suggests that rodent efficacy and safety data may underestimate human neurodevelopmental vulnerability. Therefore, confirmatory studies in higher-order systems, such as human iPSC-derived organoids, humanized animal models, or non-human primates, are critical for improving clinical translation. Together, these complementary models will help ensure that early-phase exploratory trials remain relevant to the human disease trajectory.

Comments:

  1. Figures and Tables
    • Figure 1 and Figure 2 are useful but would benefit from more explanatory legends (e.g., acronyms like ENGase, NRF1 should be spelled out).
    • Table 1: Include approximate patient numbers or references for prevalence rates to give a stronger clinical context.
    • Table 2: Add a column for current development stage (preclinical, early-phase clinical, etc.) to give readers a sense of progress.

Responses: Thank the reviewer for the advice.

We have updated the legends for Figures 1 and 2 to provide clearer explanations.

For Table 1, we added approximate patient numbers and prevalence rates, along with citations for the sources of these data.

For Table 2, we included a “Current Development Stage” column (preclinical, early-phase clinical, etc.) to give readers a sense of progress.

Minor Comments

  1. Writing and Clarity

        The manuscript is generally well written, but several sentences are lengthy and could be tightened for readability.

        Example: Lines 21–24 (“We discuss the molecular basis … alleviate neurological symptoms”) could be split for clarity.

Responses: Thank the reviewer for pointing this out. We have revised this sentence in the text as follows (Lines 21-24): We discuss the molecular basis of NGLY1 deficiency in the central nervous system (CNS) and pathophysiological insights from animal and human induced pluripotent stem cell (iPSC)-based models. We also highlight emerging gene therapy approaches aimed at restoring NGLY1 activity and alleviating neurological symptoms.

Comments:

  1. References

         Ensure reference formatting is consistent (some entries miss volume/issue formatting).

         More recent papers (2023–2025) on clinical trial designs in rare genetic disorders could be cited for context.

Responses: Thank the reviewer for the advice. We carefully checked the references and made sure that the formatting is consistent throughout the manuscript. In addition, we have cited reviews on recent clinical trial development for rare diseases and added these information to Section 4.

Clinical trials for rare genetic disorders have advanced significantly, particularly in gene therapy and personalized medicine, though challenges in accessibility and regulatory pathways remain. Advances in genomics, including next-generation and whole-genome sequencing, have accelerated both diagnosis and treatment development (Baylot et al, 2024; Videnovic et al, 2023). Recent trials increasingly incorporate CRISPR-Cas9 gene editing, RNA-based therapies, and small molecule drugs. In 2025, the U.S. FDA introduced Rare Disease Evidence Principles to expedite approvals for ultra-rare genetic conditions, emphasizing flexibility in trial design given small patient populations. Complementary efforts, such as the Ultra-rare Gene-based Therapy Network, provide funding and infrastructure to advance gene therapy research for ultra-rare neurological disorders. The clinical trial development of NGLY1 deficiency will benefit from these resources, as they provide not only funding and infrastructure but also access to specialized expertise, standardized protocols, and regulatory guidance. Leveraging these initiatives can help overcome barriers posed by small patient populations, accelerate the transition from preclinical findings to human trials, and ensure that study designs are both feasible and scientifically rigorous. Together, these resources create an ecosystem that supports the advancement of safe, effective, and patient-centered therapies for NGLY1 deficiency.

Comments:

  1. Conflict of Interest

          One author is employed by a company (OrganoidPro LLC). While disclosed, it may be appropriate to clarify whether this affiliation had any role in data interpretation, especially regarding organoid models.

Responses: Thank the reviewer for the reminder. Although Dr. YC Wang is currently employed by OrganoidPro LLC, we would like to state the fact that the research/topics discussed in the current manuscript has no financial/commercial relationship with Dr. Wang’s company affiliation and that there is no conflict of interest from his side in this manuscript. Dr. Wang’s academic laboratory has worked on NGLY1 research in the past decade; the operation of this research program is independent of Dr. Wang’s duty in the current company.

Comments:

  1. Terminology

          Use consistent terminology: “NGLY1 deficiency” vs. “NGLY1-CDDG.” Consider unifying to avoid reader confusion.

Responses: Thank the reviewer for pointing this out. In Section 1 “Introduction” of our manuscript, we clarified that NGLY1-CDDG is synonymous with NGLY1 deficiency and have used the term NGLY1 deficiency consistently throughout the text.

Reviewer 2 Report

Comments and Suggestions for Authors

The manuscript entitled “NGLY1 as an emerging critical modulator for neurodevelopment and pathogenesis in the brain” is a well-written and comprehensive review that addresses an important and relatively underexplored topic in molecular neuroscience. The authors summarize the current knowledge about NGLY1 biology, its role in neural development, and its implication in NGLY1 deficiency, integrating findings from animal models, patient-derived iPSCs, and organoid systems. The text is clear, well-structured, and accessible, making it suitable for the readership of IJMS.

 The manuscript is logically organized, progressing from molecular mechanisms to clinical aspects and therapeutic strategies. The review covers enzymatic and non-enzymatic functions of NGLY1, neural phenotypes in animal and human models, and therapeutic avenues including gene therapy and CRISPR-based approaches.  With the rapid expansion of rare disease research and gene therapy, this review provides valuable synthesis for both basic scientists and clinicians.

Points for improvement:

  1. Depth of critical discussion – While the review is descriptive, in some sections (e.g., organoid-based findings, immune signaling pathways), the text could benefit from a more critical evaluation of limitations and open questions rather than mainly summarizing existing work.

  2. Clinical translation – The therapeutic strategies section could be expanded with a more balanced discussion on challenges (e.g., safety, delivery methods, regulatory questions), to temper the optimism with realistic perspectives.

  3. Minor redundancies – Certain mechanistic details (e.g., ERAD-related pathways) are described in both the introduction and body of the text. These could be slightly condensed for readability.

  4. References – The reference list is extensive, but a few very recent publications on organoid modeling and biomarker development might be more explicitly highlighted in the conclusion to underscore future direction

Author Response

Reviewer 2

The manuscript entitled “NGLY1 as an emerging critical modulator for neurodevelopment and pathogenesis in the brain” is a well-written and comprehensive review that addresses an important and relatively underexplored topic in molecular neuroscience. The authors summarize the current knowledge about NGLY1 biology, its role in neural development, and its implication in NGLY1 deficiency, integrating findings from animal models, patient-derived iPSCs, and organoid systems. The text is clear, well-structured, and accessible, making it suitable for the readership of IJMS.

 The manuscript is logically organized, progressing from molecular mechanisms to clinical aspects and therapeutic strategies. The review covers enzymatic and non-enzymatic functions of NGLY1, neural phenotypes in animal and human models, and therapeutic avenues including gene therapy and CRISPR-based approaches.  With the rapid expansion of rare disease research and gene therapy, this review provides valuable synthesis for both basic scientists and clinicians.

Points for improvement:

Comments:

  1. Depth of critical discussion – While the review is descriptive, in some sections (e.g., organoid-based findings, immune signaling pathways), the text could benefit from a more critical evaluation of limitations and open questions rather than mainly summarizing existing work.

Responses: Thank the reviewer for the advice. We have added the following to Section 3.6 to strengthen the discussion on limitations and open questions on organoids-based findings and immune signaling).

Modeling NGLY1 deficiency with hiPSC-derived COs offers both advantages and limitations. COs reveal core NGLY1 phenotypes, including premature neurogenesis, loss of SATB2+ upper-layer neurons, and disruption of STAT3/HES1, with down-regulation of IGFBP2/ID4. Patient-derived midbrain organoids also show reduced dopaminergic markers. These models are stress-sensitive (proteasome inhibition, oxidative and ER stress) and can be rescued by NGLY1 re-expression. Critically, however, standard or-ganoids model early fetal stages and lack full vascular and immune components, limiting inferences about later maturation and neuroinflammation. Priority next steps include testing ENGase-dependent N-GlcNAc aggregation and its reversibility (ENGase inhibition) in neural contexts, moving to vascularized organoids and micro-glia-containing assembloids, extending culture time, and aligning organoid readouts with translational biomarkers such as GlcNAc-Asn.

For immune signaling, current evidence shows that NGLY1 loss activates cGAS–STING/MDA5–MAVS and interferon programs in fibroblasts via impaired NRF1-linked proteostasis/mitochondria, whereas neural cells in cerebral organoids do not display the same IFN activation at baseline, a gap likely reflecting cell-type thresholds and missing glia/immune crosstalk. Moreover, NGLY1 promotes PD-1 deglycosylation and degradation in T cells (without destabilizing PD-L1), suggesting context-dependent immunomodulation whose CNS relevance remains to be tested. A more mechanistic program should probe triggered conditions (e.g., proteasome/oxidative stress where organoids are vulnerable), quantify innate-immunity readouts (cGAMP, STING phosphorylation, ISG proteins), and ask whether NRF1 restoration or proteostasis modulators normalize mitochondrial/IFN phenotypes, ideally in neuron–microglia (with or without T-cell) co-cultures that can reveal PD-1/PD-L1 pathway consequences in neural circuits.

Comments:

  1. Clinical translation – The therapeutic strategies section could be expanded with a more balanced discussion on challenges (e.g., safety, delivery methods, regulatory questions), to temper the optimism with realistic perspectives.

Responses: Thank the reviewer for the advice. We have added the following to Section 4 to emphasize on challenges and realistic perspectives.

Proof-of-concept for AAV9 gene replacement and CRISPR/Cas9 correction is strong in rodents and patient-derived cells, with restoration of NGLY1 activity and functional improvement after CNS-directed or systemic AAV9 delivery, as well as successful gene correction in iPSC models. Nonetheless, clinical translation will require caution in three key areas: safety, delivery, and durability. Safety considerations include vector-related toxicity, potential immune activation, and off-target gene-editing effects. For delivery, the challenge is to achieve broad and sustained expression across heterogeneous brain regions, while preclinical work must carefully weigh trade-offs between intracerebroventricular/intrathecal and systemic routes. With respect to durability and re-dosing, strategies must ensure long-term expression in the context of a lifelong disorder. From a regulatory perspective, progress will depend on (i) rigorous biodistribution and potency data in relevant CNS models, (ii) trial designs that prioritize early intervention, and (iii) incorporation of surrogate or response biomarkers, particularly GlcNAc-Asn, to monitor pharmacodynamic effects in small patient cohorts. In parallel, a complementary small-molecule strategy should be advanced, using iPSC-based screening to identify agents that modulate NRF1, enhance proteostasis, or mitigate cellular stress. Such agents could be developed as monotherapies or as adjuncts to gene therapy, diversifying risk while gene-based approaches mature. Taken together, this tempered outlook balances optimism with pragmatic steps to accelerate translation, reduce risk, and ensure that forthcoming trials are interpretable and clinically meaningful.

Comments:

  1. Minor redundancies – Certain mechanistic details (e.g., ERAD-related pathways) are described in both the introduction and body of the text. These could be slightly condensed for readability.

Responses: Thank the reviewer for the advice. We have revised the mechanism on ERAD related pathways as follows to make it more concise in Section 2.1.

As a cytoplasmic PNGase, NGLY1 has been reported to deglycosylate several ERAD substrates, although whether Ngly1-mediated deglycosylation plays an essential role in ERAD substrate degradation remains undetermined (Wiertz et al., 1996; Suzuki et al., 1997; Mosse et al., 1998). For example, although downregulation of Ngly1 reduces the deglycosylation of the TCR α subunit and MHC class I heavy chain, two well-known glycoproteins and ERAD substrates, their proteasome-mediated degradation was not affected (Hirsch et al., 2003; Blom et al., 2004; Misaghi et al., 2004). In contrast, the degradation of EDEM1, a glycoprotein that targets misfolded proteins in the ER for degradation, appeared to be Ngly1-dependent and disrupted in response to Ngly1 inhibition (Park et al., 2014). Moreover, RTAΔm, a model ERAD substrate, accumulates in Ngly1-knockout mouse embryonic fibroblasts (Huang et al., 2015), indicating the critical role of NGLY1 on degradation of RTAΔm. Overall, NGLY1 facilitates ERAD by accelerating the turnover of a subset of misfolded glycoproteins, but ERAD is not universally dependent on NGLY1, as several substrates (e.g., TCRα, MHC-I) are still degraded when NGLY1 is reduced, whereas others (e.g., EDEM1, RTAΔm) show clearer NGLY1 dependence or compensation by cytosolic ENGase, a cytosolic endo-β-N-acetylglucosaminidase, which can generate N-GlcNAc–modified proteins prone to aggregation.

Comments:

  1. References – The reference list is extensive, but a few very recent publications on organoid modeling and biomarker development might be more explicitly highlighted in the conclusion to underscore future direction

Responses: Thank the reviewer for the advice. We have added the following to Section 5 “conclusion” to highlight on future direction.

Importantly, hiPSC-derived derived brain organoids provide a powerful platform to study how NGLY1 deficiency disrupts neurogenesis, revealing alterations in neural stem cell signaling, premature differentiation, and loss of key neuronal populations. Collectively, organoid studies underscore the critical role of NGLY1 in human brain maturation and establish hiPSC-derived systems as indispensable for mechanistic and therapeutic exploration (Yang et al., 2019; Du et al., 2021; Pradhan et al., 2021; Lin et al., 2022; Sasserath et al., 2022; Abbott et al., 2023; Manole et al., 2023; Shyr et al., 2025)

Reviewer 3 Report

Comments and Suggestions for Authors

Review Zhang et al

Zhan et al. present a short review about the current knowledge on NGLY1, the cytoplasmic enzyme that removes N-linked glycans from retrotranslocated proteins from the ER. The review is structured in introduction, NGLY1 function in the CNS, CNS-pathologies caused by NGLY1 malfunction, therapeutic strategies and conclusions. The manuscript is well written and the figures are nice and clear. I only have some suggestions that could further improve the manuscript.

  1. NGLY1 is the name of the gene whereas, according to Uniprot, N-Glycanase 1 or PNGase or hPNGase are the protein names. I suggest you decide for a single protein name and stick to it consequently throughout the manuscript when you talk about the protein. When you talk about the gene use NGLY1 in italics.

  1. Last sentence in 2.1 “Overall….” and Fig. 1. It seems to me that this last sentence of the paragraph and the figure is a bit at odds with what you write in line 103 ff, that in fact it is not clear whether PNGase is essential for ERAD or not. From your text I take that it may rather be involved in the degradation for a specific set of substrates only, since ERAD works also in the absence of PNGase.

  1. Sentences 1-3 in 3.3. In the first sentence you state as fact that NGLY1 contributes to other neuropathological conditions, but then you say additional studies…if and how…maybe involved. So it is not clear at all whether it is involved in AD, PD, etc, right?

  1. Table 2 and paragraph 4. Text and table to me appear not well connected, for example the iPSC-induced drug screening is not referred to at all in the text. I suggest to revise that paragraph.

  1. The conclusion is a bit superficial and superfluous. I suggest to also emphasize what is not clear, for example which are the important substrates of PNGase that contribute to the pathology when not deglycosylated, what is the impact of the non-enzymatic function of PNGase on the disease, are there for example mutations that disrupt enzymatic function but not stability?

Author Response

Reviewer 3

Zhang et al. present a short review about the current knowledge on NGLY1, the cytoplasmic enzyme that removes N-linked glycans from retrotranslocated proteins from the ER. The review is structured in introduction, NGLY1 function in the CNS, CNS-pathologies caused by NGLY1 malfunction, therapeutic strategies, and conclusions. The manuscript is well written, and the figures are nice and clear. I only have some suggestions that could further improve the manuscript.

Comments:

  1. NGLY1 is the name of the gene whereas, according to Uniprot, N-Glycanase 1 or PNGase or hPNGase are the protein names. I suggest you decide for a single protein name and stick to it consequently throughout the manuscript when you talk about the protein. When you talk about the gene use NGLY1 in italics.

Responses: Thank the reviewer for pointing this out. We have made the nomenclature consistent throughout the text. For clarity and consistency, we defined the nomenclature in the first sentence of Section 1 “Introduction”: N-Glycanase 1 (also known as PNGase or human PNGase, hereafter referred to as NGLY1 for the protein, NGLY1 (in italic) for the human gene, and Ngly1 (in italic) for the rodent gene).

Comments:

  1. Last sentence in 2.1 “Overall….” and Fig. 1. It seems to me that this last sentence of the paragraph and the figure is a bit at odds with what you write in line 103 ff, that in fact it is not clear whether PNGase is essential for ERAD or not. From your text I take that it may rather be involved in the degradation for a specific set of substrates only, since ERAD works also in the absence of PNGase.

Responses: Thank the reviewer for pointing this out. We have revised the last sentence in Section 2.1. as follows:

Overall, NGLY1 facilitates ERAD by accelerating the turnover of a subset of misfolded glycoproteins, but ERAD is not universally dependent on NGLY1: several substrates (e.g., TCRα, MHC-I) are still degraded when NGLY1 is reduced, whereas others (e.g., EDEM1, RTAΔm) show clearer NGLY1 dependence or compensation by cytosolic ENGase, which can generate N-GlcNAc–modified proteins prone to aggregation.

In addition, we have added the following to Figure 1 legend for clarity: ERAD can proceed via NGLY1-independent routes for certain substrates (dashed path via ENGase), while other substrates exhibit partial NGLY1 dependence; thus, NGLY1 acts as a fa-cilitator rather than a universal requirement for ERAD. ENGase, en-do-beta-N-acetylglucosaminidase, a cytosolic enzyme that hydrolyzes mannose-modified pep-tides and proteins, generating free oligosaccharides. NRF1, nuclear respiratory factor 1, a tran-scription factor regulating mitochondrial function, proteasome activity, and stress responses, essential for cellular homeostasis and adaptation.

Comments:

  1. Sentences 1-3 in 3.3. In the first sentence you state as fact that NGLY1 contributes to other neuropathological conditions, but then you say additional studies…if and how…maybe involved. So it is not clear at all whether it is involved in AD, PD, etc, right?

Responses: Thank the reviewer for pointing this out and we apologize for the confusion. We have revised this part as follows:

Beyond neurodevelopmental defects, NGLY1 dysfunction may also influence neurodegenerative processes. Recent work suggests that NGLY1 regulates tau seeding, aggregation, and turnover (Batra et al, 2024), raising the possibility of a broader role in disorders such as Alzheimer’s disease, Parkinson’s disease, amyotrophic lateral sclerosis, frontotemporal dementia, and Huntington’s disease, all of which share proteotoxic stress as a hallmark. However, no causal link between NGLY1 and these conditions has been established. Instead, current findings highlight a potential mechanistic connection. For example, the reported role of NGLY1 in tau regulation could position it as a possible modifier of neurodegenerative pathways, pending further validation in disease models and patient tissues (Ruz et al, 2020).

Comments:

  1. Table 2 and paragraph 4. Text and table to me appear not well connected, for example the iPSC-induced drug screening is not referred to at all in the text. I suggest to revise that paragraph.

Responses: Thank the reviewer for the advice. We have revised both Table 2 and Section 4 to connect better and to include iPSC-based drug screening.

Comments:

  1. The conclusion is a bit superficial and superfluous. I suggest to also emphasize what is not clear, for example which are the important substrates of PNGase that contribute to the pathology when not deglycosylated, what is the impact of the non-enzymatic function of PNGase on the disease, are there for example mutations that disrupt enzymatic function but not stability?

Responses: Thank the reviewer for the advice. We have now added the following to Section 5 “Conclusion” to provide. potential mechanistic insight.

NGLY1 deficiency clearly perturbs proteostasis and neurodevelopment, but decisive CNS mechanisms remain unresolved. ERAD is not universally dependent on NGLY1, as some substrates are degraded without it, whereas others show clearer NGLY1 dependence or ENGase compensation that produces N-GlcNAc aggregates. Beyond catalysis, NGLY1 exerts non-enzymatic functions, including aquaporin transcription (rescued by a catalytic-dead allele) and regulation of NRF1-driven proteasome/mitochondrial programs. How these roles partition across neural lineages is unclear. With >70 pathogenic variants and marked phenotypic heterogeneity, potential separation-of-function classes are possible but unmapped. Priorities now are cell-type-resolved N-glycoproteomics in human neural models, genetic dissection of the NGLY1-ENGase axis, allele-specific rescue (catalytic-dead vs. wild-type) to parse catalytic vs regulatory roles, and structure-guided variant stratification in isogenic neural cells linked to patient phenotypes; these advances will guide when to prioritize enzyme restoration versus proteostasis or NRF1 modulation and will enable variant-informed biomarkers (e.g., GlcNAc-Asn) in clinical trials.